# Brain–Computer Interface Based on Steady-State Visual Evoked Potential Using Quick-Response Code Pattern for Wheelchair Control

**DOI:** 10.3390/s23042069

**Published:** 2023-02-12

**Authors:** Nannaphat Siribunyaphat, Yunyong Punsawad

**Affiliations:** 1School of Informatics, Walailak University, Nakhon Si Thammarat 80160, Thailand; 2Informatics Innovative Center of Excellence, Walailak University, Nakhon Si Thammarat 80160, Thailand

**Keywords:** brain–computer interface, steady-state visual evoked potential, quick response, QR code, wheelchair

## Abstract

Brain–computer interfaces (BCIs) are widely utilized in control applications for people with severe physical disabilities. Several researchers have aimed to develop practical brain-controlled wheelchairs. An existing electroencephalogram (EEG)-based BCI based on steady-state visually evoked potential (SSVEP) was developed for device control. This study utilized a quick-response (QR) code visual stimulus pattern for a robust existing system. Four commands were generated using the proposed visual stimulation pattern with four flickering frequencies. Moreover, we employed a relative power spectrum density (PSD) method for the SSVEP feature extraction and compared it with an absolute PSD method. We designed experiments to verify the efficiency of the proposed system. The results revealed that the proposed SSVEP method and algorithm yielded an average classification accuracy of approximately 92% in real-time processing. For the wheelchair simulated via independent-based control, the proposed BCI control required approximately five-fold more time than the keyboard control for real-time control. The proposed SSVEP method using a QR code pattern can be used for BCI-based wheelchair control. However, it suffers from visual fatigue owing to long-time continuous control. We will verify and enhance the proposed system for wheelchair control in people with severe physical disabilities.

## 1. Introduction

Recently, brain–computer interfaces have been intensely developed in terms of hardware and software for practical use [1]. In particular, brain signal acquisition technology has demonstrated a new technique with high flexibility and usability [2]. In addition, invasive BCI research [3,4] has shown straightforward advances in medicine, which can lead to innovations in BCI that are important for human beings in the future, together with other modern sciences, such as artificial intelligence and robotics. An electroencephalogram-based BCI is a non-invasive technique [5] that receives brain signals from the scalp to measure the electrical potential of the brain and converts it into commands to control electrical devices, computers, or machines without motor functions [6,7]. BCI systems are used in medical applications [8], such as diagnosis [9,10,11,12], treatment [13,14], and rehabilitation [15,16,17,18,19].

Physical disabilities significantly affect daily activities, especially for those with severe disabilities who cannot help themselves. The development of assistive systems and tools to facilitate or rehabilitate people with disabilities has become more widespread. Brain-controlled systems can be alternative solutions for severe disabilities [20,21,22]. BCI systems’ research and development aim to make their daily lives as close to normal as possible. BCI-based practical machine control is famous for enhancing the movement abilities of people with physical disabilities. Utilizing BCI for external devices to control a robot hand or exoskeleton can recover essential movement functions of the disabled or the elderly [23,24,25,26]. Cooperation between BCI and intelligent systems or smart devices employing artificial intelligence can suggest highly efficient practical approaches.

A wheelchair is an assistive mobility device that can increase the workspace for performing activities. An EEG-based BCI for powered wheelchair control [27,28,29] is an alternative system for severe movement disabilities. The popular EEG technique for BCI-based machine control consists of motor imagery (MI) [30], visual stimulation [31], or combining both techniques [32]. EEG features from visual stimulation are called visual evoked potentials (VEPs), which can be divided into two types: a transient visual evoked potential (P300) and steady-state visual evoked potential. The use of BCI for wheelchair control is summarized in Table 1. Many researchers have been interested in developing BCI systems to control the movement of electric wheelchairs to help patients. Several techniques have been applied, including the motor imagery method. For example, Xiong et al. [33] developed a system using MI to imagine raising the left and right hands with a jaw clenched for testing with an actual wheelchair. The results revealed a mean accuracy of 60 ± 5%, whereas the participants could achieve results with an accuracy of 82 ± 3%. Permana et al. [34] used MI with eye movement to control a wheelchair by imagining moving forward and backward as well as thinking forward and backward with eye movement to control left and right turns. The correct value was in the range of 64–82.22%. The P300 method is another technique that is used for wheelchair control. Eidel et al. [35] developed a wheelchair-control system using tactile stimulation of the legs, abdomen, and neck. It was tested in a simulated environment to study the offline and online results. The offline results showed an average accuracy of 95%, whereas the accuracy of the online control was 86%. Chen et al. [36] used the P300 method to control an actual wheelchair using a control panel with a micro-projector to create a blinking visual stimulus. The results yielded an average accuracy of 88.2%. In addition, the MI method, in combination with the P300 method, was used to control the wheelchair’s movement. For example, Yu et al. [37] used flashed arrows to control the direction and imagined raising the left and right hands to adjust the wheelchair’s speed. The results showed that the MI method was 87.2% accurate, and the accuracy of using P300 for controlling the wheelchair was 92.6%. Furthermore, several researchers are interested in using the SSVEP method to develop wheelchair motion control. For example, Chen et al. [38] designed arrow patterns with different flicker frequencies to control an actual wheelchair by displaying the stimulus on regular monitors and mixed reality (MR) goggles. This study employed canonical correlation analysis (CCA) and microsatellite instability (MSI) analysis methods to determine their accuracy. The stimulus displayed via the MR goggles using CCA for analysis yielded a maximum accuracy of 98.8%. Na et al. [39] used an LCD monitor with an LED metric as a backlight to display an SSVEP stimulus for driving a wheelchair. This method yielded an average accuracy of 93.9%. Ruhunage et al. [40] developed an SSVEP stimulus combined with an Electrooculogram (EOG) to control wheelchairs and other home devices. They used visually stimulating LEDs to control wheelchairs and to change modes. Moreover, blinking twice stopped the wheelchair command. The mean accuracy of SSVEP was 84.5%, and that of the eye blinking was 100%. In addition, the steady-state motion visual evoked potential (SSMVEP) method [41] can be employed to enhance SSVEP-controlled wheelchairs during visual fatigue. The SSMVEP can yield an average accuracy of 85.6%. However, SSVEP can achieve an accuracy higher than the SSMVEP method.

According to Table 1, the SSVEP is a popular method for EEG-based BCI. The advantage is that it requires less training time than other methods and has high accuracy [42]. However, the SSVEP method requires a flicker stimulus pattern with high illuminance to yield an apparent response at the flickering frequency; however, this can easily lead to visual fatigue. Therefore, an SSVEP stimulus pattern was developed for a practical SSVEP-based BCI system. For example, Keihani et al. [43] investigated impulse improvement by modulating the sinusoidal wave frequency of a stimulus to reduce visual acuity. Visual acuity could be reduced to a small percentage of conventional stimuli and had an accuracy of 88.35% in power spectral density (PSD) detection and 90% in CCA, as well as in the least absolute shrinkage and selection operator (LASSO) analysis. Moreover, a study of the effects of color and stimulation frequency in SSVEP [44] was developed by Duart et al. using white, red, and green colors in combination with three frequencies of 5 Hz, 12 Hz (medium), and 30 Hz (high), with spectral analysis from the measurements. The signal-to-noise ratio (SNR) obtained from an analysis of variance (ANOVA) at the mid-frequency yielded the best SNR, whereas the white and red colors produced good SNR. Furthermore, Mu et al. proposed frequency superposition to superimpose different frequencies using the OR method with ADD to make the LED flicker [45]. The results showed that the average accuracy of the improved method with CCA for multi-frequency CCA (MFCCA) was higher than that of the conventional pattern. The design of novel stimulus patterns can reduce visual fatigue and maintain the system’s performance.

This paper describes the development of an SSVEP-based BCI system for simulated wheelchair control as a prototype before actual wheelchair implementation. Two main issues were addressed. (1) We evaluated the efficiency of real-time SSVEP-BCI using the proposed stimulus pattern via a quick response (QR) code pattern [46]. (2) We proposed a simple algorithm for SSVEP detection. A simulated wheelchair [47] was utilized to observe the possibility of using it in an actual wheelchair.

## 2. Materials and Methods

This study demonstrated an SSVEP-based BCI system using an EEG neuroheadset for simulated wheelchair control. An overview of the wheelchair control system is shown in Figure 1. The system consists of four main parts: visual stimulation, EEG signal acquisition, feature extraction, classification algorithms, and application. The SSVEP method employs brain signal oscillations at a specific flicker frequency in the occipital area (O1 and O2). Four different frequencies were used to generate four commands of directions for wheelchair control: forward, backward, left, and right. We designed two experiments to collect and analyze data to verify the efficiency of the proposed QR code visual stimulus pattern in real-time processing using a simulated wheelchair.

### 2.1. SSVEP Stimulus Pattern

SSVEP-based BCIs suffer from visual fatigue, which can reduce the efficiency of real-time control. A novel visual stimulus pattern has been proposed for SSVEP responses that decreases visual fatigue and stimulus time [48,49,50]. Based on previous work [46], we observed the SSVEP response via a novel SSVEP stimulus pattern employing a QR code pattern (Figure 2a) using PSD and CCA methods in offline testing. The results showed that using the QR code pattern with a low frequency can yield better accuracy than the checkerboard pattern (Figure 2b) (traditional). Moreover, we found that the flicker pattern mixing the fundamental frequency and the first harmonic frequency can elicit a more robust SSVEP response than the flicker pattern with only the fundamental frequency. In addition, QR code patterns result in low eye irritation and slow visual fatigue. Therefore, this study employed a QR code pattern with a single fundamental frequency and a mixture of fundamental and first harmonic frequencies for a real-time SSVEP-based BCI system. Four flicker frequencies, 5, 6, 7, and 8 Hz, with their first harmonic, were used to generate four command controls for wheelchair steering, as shown in Table 2.

### 2.2. EEG Acquisition

In this study, we used the EPOC Flex EEG neuroheadset (Figure 3) from EMOTIV (https://www.emotiv.com, accessed on 8 September 2022) at a sampling rate of 128 Hz. The EPOC Flex is a wireless EEG machine with flexible traditional EEG head-cap systems that minimize the setup time. It measures electrical brain potentials via saline electrodes and saline-soaked felt pads. It is flexible and practical. We checked and adjusted each electrode’s position to the correct position. This device sends a signal via Bluetooth to the computer. The Cortex API is the interface between the device and the programming application. The single-channel EEG signals from channel O1 were streamed via WebSocket and then converted into JSON format with a sampling rate of 128 Hz. In preprocessing, the streamed signal was filtered using a 50 Hz notch filter for power line noise removal. A 2.5–35 Hz bandpass filter was used to remove dc noise and motion artifacts.

The data were processed using the Python programming language (version. 3.9.10) to identify the results of this research. The output command was used to control simulated wheelchairs (miWE) [51]. We used a notebook with 12 GB of RAM and a 2.3 GHz processor, and computing hardware with AMD Quad-core Ryzen 7 3700U CPU running 64-bit Windows 11 Pro. The components of the proposed SSVEP-based BCI system are illustrated in Figure 4.

Twelve healthy participants, seven females and five males (age: 27.6 ± 2.3 years old), who had been tested in previous studies, were tested. The participants did not experience color blindness or neurological disorders (in the past and present). Moreover, they did not have migraines or other symptoms affecting their eyesight. The participants read and signed a consent form for the test. This document would keep personal information confidential and anonymous. The Walailak University’s Office approved the procedure in human research of the human research ethics committee, which has endorsed the ethical declarations of Helsinki, the Council for International Organizations of Medical Sciences (CIOMS), and the World Health Organization (WHO) (protocol code: WU-EC-IN-2-076-64).

### 2.3. Proposed Algorithms

This study focused on the occipital cortex, which was activated by visual perception, to determine the relationship between target stimuli and brain-signal modulation. Based on SSVEP characteristics, frequency-domain analysis usually uses the power spectrum at specific frequencies of the EEG from flicker stimulus patterns for feature extraction [52,53]. Moreover, the power spectrum distribution can provide a robust SSVEP feature. Absolute and relative power spectral density methods were adopted to verify the proposed OR code visual stimulus pattern in real-time processing.

#### 2.3.1. Power Spectral Density

PSD is a common signal-processing technique that distributes the signal power over a frequency and shows the magnitude of energy as a function of frequency [52]. The input signal was converted from the time domain to the frequency domain to determine the PSD value. The segmented signal was separated into overlapping segments at the same window length, which reduces the variance of the periodogram. The output frequency components were calculated using a fast Fourier transform (FFT)-based periodogram algorithm. The Welch periodogram can estimate the PSD by averaging the squares of the spectral density for each interval [53]. The Welch estimation of PSD can be obtained by Equation (1):(1)$$P_{w}\left(f\right)=\frac{1}{KNU}\overset{K}{\underset{k=1}{\sum }}\;{\left|\overset{K}{\underset{k=1}{\sum }}x\left(n\right).w\left(n\right)\;e^{\left(-j2\pi fn\right)}\right|}^{2}$$
where N is the length of the signal, K is the number of overlapped segments, D is the shifted points, n is the number of sequences, x(n) = x(n−D), w(n) is a window function, and f is a frequency component.  U is a constant of the power of the window function used as a normalizing factor in Equation (2), which is calculated as follows:(2)$$U=\frac{1}{N}\overset{N}{\underset{n=1}{\sum }}{\left|w\left(n\right)\right|}^{2}$$

In this study, EEG signals were collected every 4 s (N = 512). The EEG signal was windowed using Hamming windows with a 50% overlap for PSD estimation.

#### 2.3.2. Relative Power Spectral Density

The PSD can be used to perform spectral analysis, which can be extracted from the total power, relative power, and absolute power of the estimated Welch PSD values of the EEG signals. Spectral features can be used for SSVEP detection. The relative PSD is the ratio between the absolute power of the PSD at a specific frequency band and the total power of the PSD of the entire frequency band. Relative PSD can reduce inter-individual differences from the absolute power [52]. Furthermore, we found that changing one frequency band affected the difference in the relative PSD. Hence, we attempted to employ the relative PSD for the SSVEP parameters. In this approach, we adopted the relative PSD for the SSVEP features. Relative PSD values could be calculated by summing the PSD values at a specific frequency and the harmonics divided by the sum of the PSD values at all flickering frequencies and their harmonics as follows in Equation (3):(3)$$\;Relative\;P\left(f_{i}\right)=\frac{\sum \left(f_{i,}\;f_{2i,}\;f_{3i}\;\right)}{\sum \left(f_{t}\;\right)}\;$$
where $f_{i}$ is the magnitude of PSD at the target frequency, i is the flickering frequency of the stimulus (i.e., 5, 6, 7, and 8 Hz), $f_{t}$ is the total magnitude of the PSD at all flickering, first harmonic, and second harmonic frequencies.

#### 2.3.3. Feature Extraction and Decision-Making Algorithms

For the SSVEP feature extraction, we considered the PSD at the fundamental and harmonic frequencies for the feature parameters from the absolute and relative PSD algorithms. The first harmonic frequencies were 10, 12, 14, and 16 Hz. The second harmonic frequencies were 15, 18, 21, and 24 Hz.

(1)Calibrating and Threshold setting

In this approach, before starting the process, the parameters of each feature algorithm must be recorded, while the participant focuses on fixation (+) for 4 s, as follows:

A1. The maximum PSD of the fundamental frequency of the baseline (flickering frequency) and its first and second harmonics were recorded. To distinguish the SSVEP response, spontaneous EEG baseline values ($B_{i}$) represented the maximum magnitudes of PSD at the fundamental frequency, and harmonic frequencies of the EEG signal from channel O1 were collected, which could be calculated using Equation (4) to provide threshold values ($T_{i}$) as follows in Equation (5).
(4)$$B_{i}=\underset{\;}{\max}\left(f_{i,}\;f_{2i,}\;f_{3i}\;\right)$$
(5)$$T_{i}=1.5*B_{i}$$

A2. The relative PSD of the fundamental frequency of baseline values ($BR_{i}$) was collected following Equation (6) to provide threshold values ($TR_{i}$) as denoted in Equation (7).
(6)$$BR_{i}=Relative\;P\left(f_{i}\right)$$
(7)$$TR_{i}=1.5*BR_{i}$$

Next, the parameters of each feature algorithm must be recorded while the participant focuses on the flickering frequency, as follows:

B1. The maximum PSD of the fundamental frequency and its harmonics ($P_{i}$) were collected as shown in Equation (8):(8)$$P_{i}=\underset{\;}{\max}\left(f_{i,}\;f_{2i,}\;f_{3i}\;\right)$$

B2. Simultaneously, the relative PSD of the fundamental frequencies ($RP_{i}$) was collected as shown in Equation (9):(9)$$RP_{i}=Relative\;P\left(f_{i}\right)$$

(2)Feature Extraction:

The SSVEP response was higher than the baseline value. SSVEP feature extraction could be performed as follows:

C1. The absolute PSD of the SSVEP ($A_{i}$) represented the difference between $P_{i}$ and $T_{i}$ at *i* = 5, 6, 7, and 8 Hz, which can be obtained by Equation (10):(10)$$A_{i}=\{\begin{array}{c} P_{i}-T_{i}\;\;\;\;\;\;\;\;\;\;,\;\;P_{i}-T_{i}>0\\ \;\;\;\;0\;\;\;\;\;\;\;\;\;\;\;\;\;\;\;\;,\;\;P_{i}-T_{i}<0 \end{array}$$

The process output represented the index of the maximum value of $P_{out}$ that was returned from the argument max function (argmax), which can be calculated as shown in Equation (11):(11)$$P_{out}=argmax\{A_{5},\;A_{6},\;A_{7},\;A_{8}\}$$

C2. The relative PSD of SSVEP ($R_{i}$), representing the difference between $RP_{i}$ and $TR_{i}$ at i = 5, 6, 7, and 8 Hz, can be calculated using Equation (12).
(12)$$R_{i}=\{\begin{array}{c} RP_{i}-TR_{i}\;\;\;\;\;,\;\;RP_{i}-TR_{i}>0\\ \;\;\;\;\;\;\;0\;\;\;\;\;\;\;\;\;\;\;\;\;\;,\;\;RP_{i}-TR_{i}<0 \end{array}$$

As with C1, the output could be identified as the index of the maximum value of$\;R_{out}$, which can be calculated following Equation (13):(13)$$R_{out}=argmax\{RP_{5},\;RP_{6},\;RP_{7},\;RP_{8}\}$$

(3)Decision making

The stream EEG signals were compared to the collected parameters during online processing every 2 s. The output command could be generated by the proposed decision-making process as follows:
Absolute PSD (C1)DecisionsRelative PSD (C2)Decisions$\mathrm{if}\;P_{out}$ = 1,Command 1$\mathrm{if}\;R_{out}$ = 1,Command 1$\mathrm{if}\;P_{out}$ = 2,Command 2$\mathrm{if}\;R_{out}$ = 2,Command 2$\mathrm{if}\;P_{out}$ = 3,Command 3$\mathrm{if}\;R_{out}$ = 3,Command 3$\mathrm{if}\;P_{out}$ = 4,Command 4$\mathrm{if}\;R_{out}$ = 4,Command 4Otherwise, Command 5Otherwise, Command 5

### 2.4. Command Translation

After processing, the classification output was translated into the control command. This process used data from the fundamental, harmonic, and neighboring frequencies according to the specified fundamental frequency for each command: 5 Hz for turning right, 6 Hz for forward, 7 Hz for backward, and 8 Hz for turning left. The baseline value was compared with the real-time value. The commands were translated to control the direction of the simulated wheelchair, as listed in Table 3.

## 3. Experiments and Results

Each participant was seated in front of the screen in a typical light environment. The distance between the participant and screen was 60 cm. The experiments were designed to verify two issues: (1) The efficiency of the SSVEP stimulation with a QR code pattern and the proposed algorithms in real-time processing. (2) The possibility of using the proposed SSVEP-based BCI system in an actual wheelchair.

### 3.1. Efficiency of Proposed Visual Stimulus Pattern

In the experiment, the accuracy of using the QR code visual stimulus pattern (Figure 5a) with only the fundamental flicker frequency and mixing the fundamental and first harmonic frequencies (Section 2.1) was collected for analysis and comparison with the conventional checkerboard visual stimulus pattern (Figure 5b). After the participants learned how to use the system, they had training sessions for 10 min before the testing started. Each participant performed the task command sequence with eight commands per trial, the command sequences as shown in Table 4, for two trials per flicker pattern (16 commands per flicker pattern) with all 64 commands. Each stimulus pattern consists of two flicker patterns (single fundamental frequency and mixture of fundamental and first harmonic frequencies, starting with a single flicker with the fundamental frequency of the checkerboard pattern. The participant rested for 5 min before beginning the subsequent trial for 5 min. This experiment showed the following results: (1) The range of average classification accuracy for all approaches. (2) Comparison of the average classification accuracy between visual stimulus patterns and feature extraction methods. (3) Average classification accuracy for each command.

According to Table 5, the average classification accuracy of the absolute PSD method ranged from 70.8% to 85.4%. Using the QR code pattern by mixing the fundamental and harmonic flicker frequencies yields a maximum average classification accuracy of 80.6%. The average classification accuracy of the proposed relative PSD method ranged from 77.1% to 95.8%. The QR code pattern also yielded a maximum classification accuracy of 95.8% by combining the fundamental and harmonic flicker frequencies. We found that the relative PSD method with checkerboard patterns and only the fundamental and mixing flicker frequencies achieved accuracies of 82.3% and 86.1%, respectively. QR code patterns with only the fundamental and mixing flicker frequencies achieved average classification accuracies of 84.4% and 92.0%, respectively. The QR code patterns produced the highest average classification accuracy by mixing fundamental and harmonic flicker frequencies for all participants, followed by checkerboard patterns with only the fundamental and mixing flicker frequencies. For the trade-off between the accuracy and the required time to run the algorithm, we calculated and observed the information transfer rate (ITR) [54,55] to investigate the applicability of the different algorithms and methods, which have also been computed as shown in Table 5. The maximum ITR with absolute PSD was 17.0 bits per minute (bpm) by the checkerboard pattern mixing the fundamental and first harmonic frequencies, but lower than the relative PSD feature. The QR code pattern mixing the fundamental and first harmonic frequencies and the relative PSD feature achieved the maximum ITR of 26.1 bpm, higher than the checkerboard pattern of 19.5 bpm.

The average classification accuracy results in Table 5 were observed using statistical analysis, as shown in Figure 6. The paired t-test (n = 12) for the mean was used to analyze statistically significant differences between sets of visual stimulus patterns. The paired t-test with 24 observations revealed a significant difference between the checkerboard and QR code patterns using the proposed relative PSD method (*p* = 0.001 and *p* < 0.005, respectively). The paired t-test indicated no significant difference between the checkerboard and QR code visual stimulus patterns when using the absolute PSD method *(p* = 0.07; *p* > 0.05). Furthermore, we considered the efficiency of each SSVEP feature-extraction method. The paired t-test with 48 observations revealed a statistically significant difference between the absolute PSD and relative PSD methods (*p* = 0.02, *p* < 0.05). Furthermore, the proposed relative PSD power can produce a higher efficiency than the absolute PSD for the SSVEP detection.

Figure 7 shows the results of the average classification accuracy of the relative PSD features of each command with different SSVEP stimulus patterns. The average classification accuracy of the turn left command ranged from 75.0% to 88.2%, the turn right ranged from 84.7% to 95.8%, the forward ranged from 88.9% to 95.8%, and the backward ranged from 79.9% to 88.2%. Moreover, the forward and turn right commands’ average accuracy exceeded 85.0% for all visual stimulus patterns. The SSVEP stimulus using the QR code 2 pattern (mixing the fundamental and harmonic flicker frequencies) achieved the highest average classification accuracy of all the commands.

### 3.2. Performance of the Proposed SSVEP-Based BCI for Control Application

According to the results in Section 3.1, the SSVEP stimulation using the QR code pattern (Figure 5a) by mixing the fundamental and first harmonic flicker frequencies and the relative PSD was used to implement the SSVEP-based BCI system. For real-time simulated wheelchair control, the miWE application [51] was employed to test the proposed BCI system. The experimental task was divided into the keyboard and BCI control sessions. Before starting the BCI session, the participants used keyboard control with their dominant hand to steer the simulated wheelchair in the testing trial, as shown in Figure 8a. The time spent from start to stop was recorded for the baseline collection to evaluate the system and user performance for independent control applications. Thereafter, the participant used the proposed BCI system to freely control the simulated wheelchair with two trials for each route. An example of this experiment is shown in Figure 8b. In addition, we recorded the time spent from start to stop to compare with keyboard control.

Figure 9 and Figure 10 show the efficiency comparisons between the proposed BCI and keyboard control based on the time required to complete routes 1 and 2, respectively. According to the results of Route 1, the average time spent with the keyboard (hand control) for all participants ranged from 42 to 53 s, and the average time was 46 s. The average time spent with the proposed BCI ranged from 134 to 261 s, and the average time was 198 s.

In Route 2, the average time spent with the keyboard for all participants ranged from 42 to 54 s, and the average time was 48 s. The average time spent with the proposed BCI ranged from 163 s to 264 s, and the average time was 214 s.

## 4. Discussion

According to the experimental results, the following three issues are important.

The first is the proposed feature extraction and classification method. Table 5 and Figure 6 show that the proposed relative PSD method yields higher efficiency than the absolute PSD method (traditional) with the proposed classification algorithm. The second regards the SSVEP stimulation patterns. Table 5 shows that using a QR code pattern can yield a higher average classification accuracy than a checkerboard pattern (conventional) and achieve high efficiency for all flicker frequencies with command translation, as shown in Figure 6. Moreover, the QR code pattern mixing the fundamental and harmonic flicker frequencies achieved a high average classification accuracy of 92.0% (Table 5). However, the maximum classification accuracy in real-time processing is lower than that in offline processing [51]. Furthermore, a comparison with previous and existing works on real-time independent wheelchair control using SSVEP modalities [38,39] indicates that the proposed SSVEP-based BCI system can produce classification accuracy and command transfer rates close to those of wheelchair control.

The third is the proposed BCI-controlled wheelchair. The results in Table 5 show the SSVEP-based BCI system for simulated wheelchair control using the QR code pattern with four fundamental mixings of their first harmonic flicker frequencies translated to the steering commands. All the participants achieved more than 85% accuracy for each command before testing (Figure 7). We observed the time taken to complete the routes in Figure 9 and Figure 10, which shows that the BCI control modality can have a lower efficiency than the keyboard for all routes. The difference between the average time taken by the BCI and keyboard control on Route 1 was 152 s and on Route 2 it was 166 s. Considering individual participants, for participants 3 and 10 with the SSVEP experience that can yield a high efficiency for BCI control, the difference in average time required between BCI and keyboard control on Route 1 was 133 s and 92 s and on Route 2 it was 128 s and 117 s, respectively. The proposed SSVEP-based BCI (brain control) was approximately three times longer than keyboard control (hand control). Compared with MI-based BCI [56] and the face–machine interface (FMI) [57] systems for simulated wheelchair control, the proposed SSVEP-based BCI system can yield a speed rate of 9.71 cm/s (ratio between distance and the average time taken), which is higher than the MI modality of 7.11 cm/s. However, still lower than the FMI modality was 12.03 cm/s. The difference between the average speed rate of the FMI and the proposed BCI system was 2.32 cm/s.

Nevertheless, some recommendations and limitations of the proposed SSVEP-based BCI system for wheelchair control are as follows:(1).The classification method is a simple algorithm for real-time processing. Using machine learning methods for classification algorithms may improve the efficiency of the proposed relative PSD features. Furthermore, multi-channel EEG signals from the occipital area might improve SSVEP features.(2).Based on the participants’ comments regarding visual fatigue, the proposed system still suffers from visual fatigue when focusing on the QR code pattern for an extended period.(3).The proposed system requires monitoring visual fatigue to avoid low accuracy.(4).Based on the results in Section 3.1, the proposed system may yield high efficiency for electric device control applications.(5).Employing the proposed system to control an actual wheelchair in a real environment should be tested for practical use.

## 5. Conclusions

This study aimed to develop an SSVEP-based BCI system using a QR code pattern for wheelchair control. The efficiency of the proposed QR code pattern for the SSVEP method was verified and compared with that of the checkerboard pattern. The QR code pattern with mixing fundamental and first harmonic flicker frequencies could achieve high efficiency for SSVEP stimulation. Four steering commands were created using four flickering frequencies. The proposed relative PSD method yielded high efficiency for the SSVEP features and classification. The proposed SSVEP stimulation and algorithms were implemented in a real-time simulation of the wheelchair control. The results presented the average classification accuracy of the proposed SSVEP method ranging from 85.4% to 95.8%. For testing with independent-based control tasks, the real-time control results of the proposed BCI control required approximately five-fold more time than keyboard control. The proposed SSVEP-based BCI system could be used for wheelchair control. However, the proposed system still suffers from visual fatigue for continuous control over a long period. In future work, we will verify and improve the proposed method to control an actual-powered wheelchair for practical use.

## Figures and Tables

**Figure 1 sensors-23-02069-f001:**
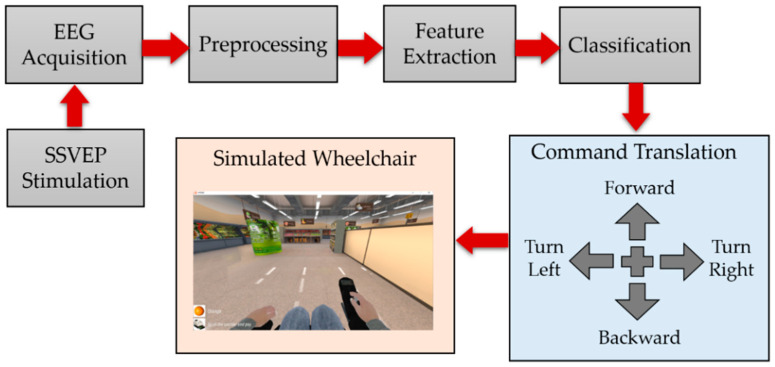
Proposed SSVEP-based BCI system using QR code visual stimulus pattern for simulated wheelchair control.

**Figure 2 sensors-23-02069-f002:**
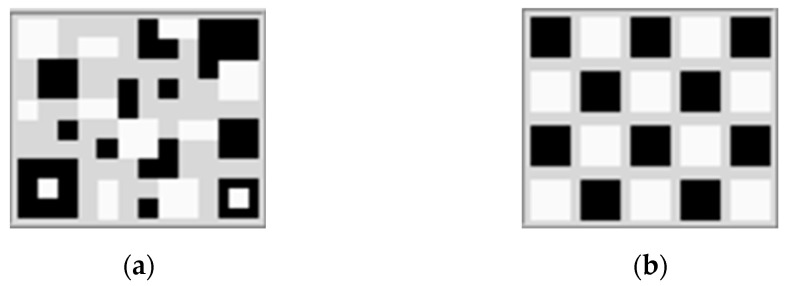
SSVEP stimulus pattern (size: 3 cm × 4 cm). (**a**) QR code pattern; (**b**) Checkerboard pattern.

**Figure 3 sensors-23-02069-f003:**
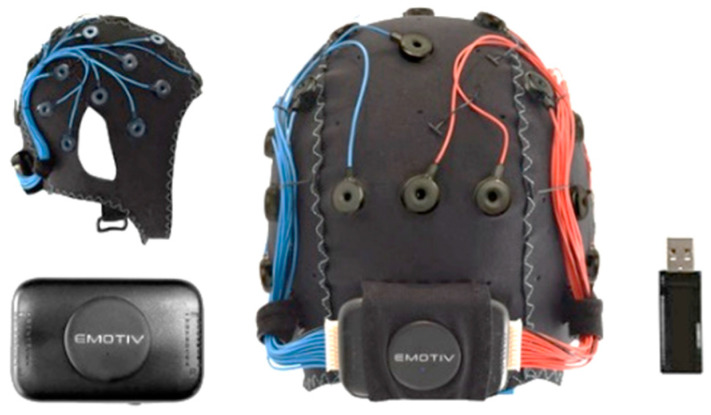
EPOC FLEX^TM^ device and accessories (https://www.emotiv.com, accessed on 8 September 2022).

**Figure 4 sensors-23-02069-f004:**
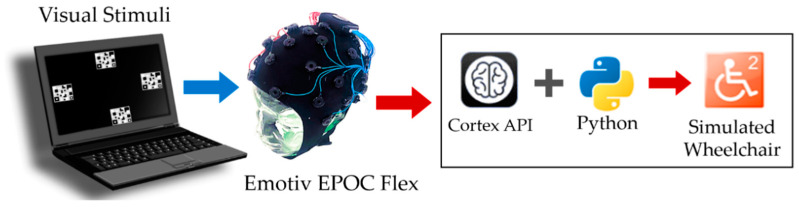
Components of the EEG acquisition using an EMOTIV EPOC Flex for a real-time BCI system.

**Figure 5 sensors-23-02069-f005:**
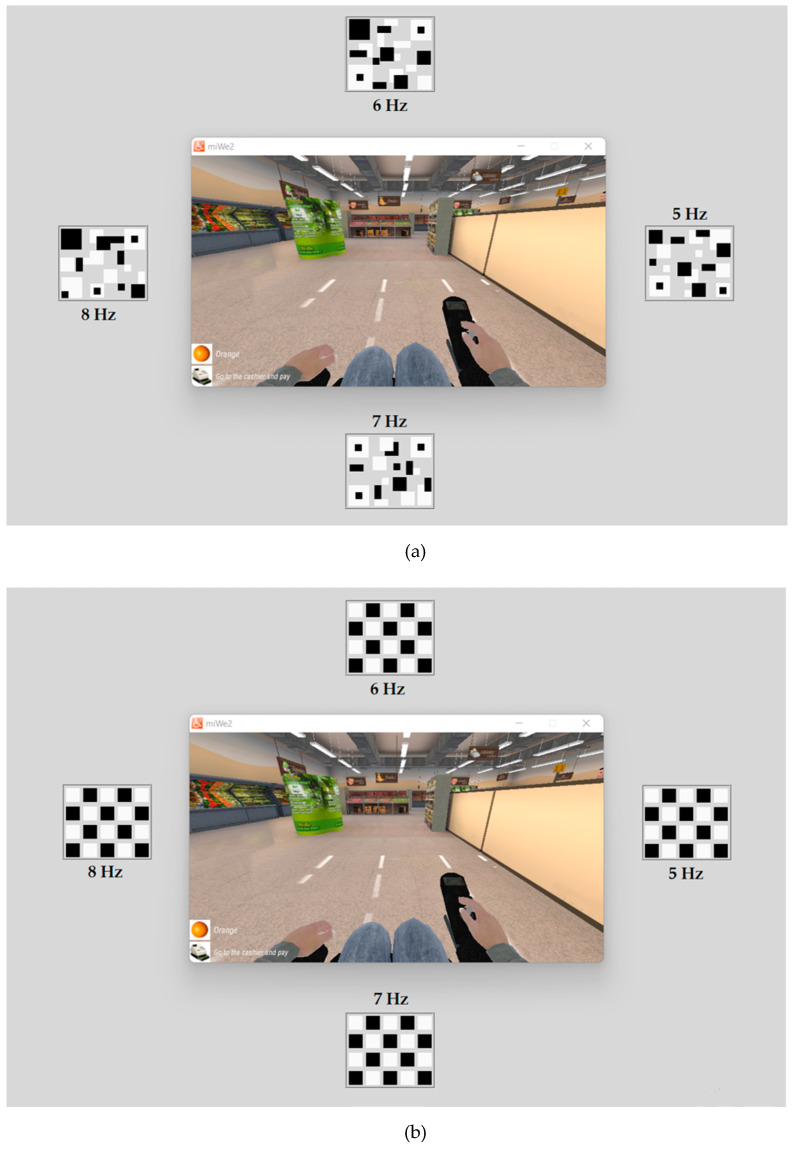
Screenshot of visual stimuli with four fundamental flicker frequencies of four BCI commands (Table 2 and Table 3) to control the simulated wheelchair through an LCD monitor. (**a**) QR code pattern (proposed); (**b**) Checkerboard pattern (traditional).

**Figure 6 sensors-23-02069-f006:**
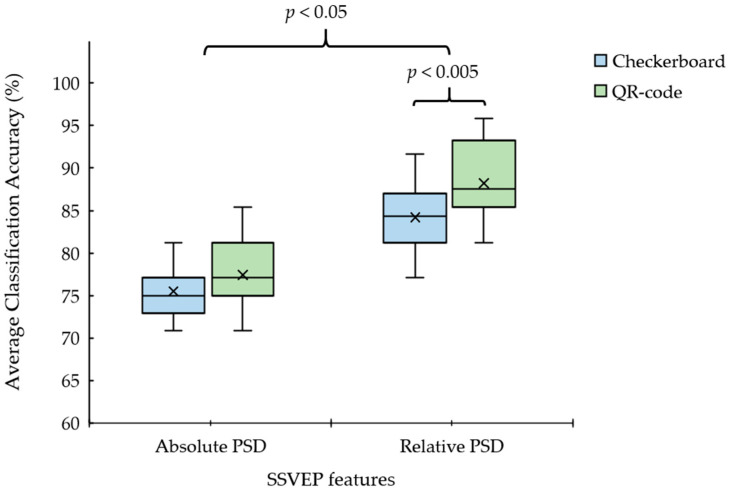
Average classification accuracy between absolute PSD and relative PSD power for SSVEP features from QR code and checkerboard stimulus patterns (shown in Table 4 and Table 5).

**Figure 7 sensors-23-02069-f007:**
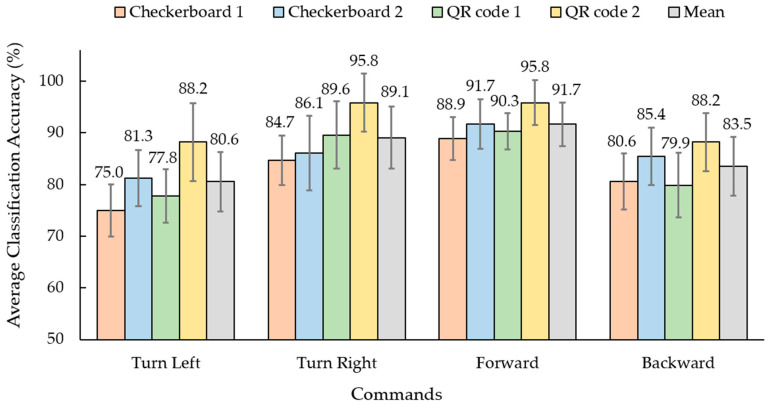
Average classification accuracy of each steering command between checkerboard and QR patterns using relative PSD method (1: only fundamental flicker frequency and 2: mixing fundamental and first harmonic frequency).

**Figure 8 sensors-23-02069-f008:**
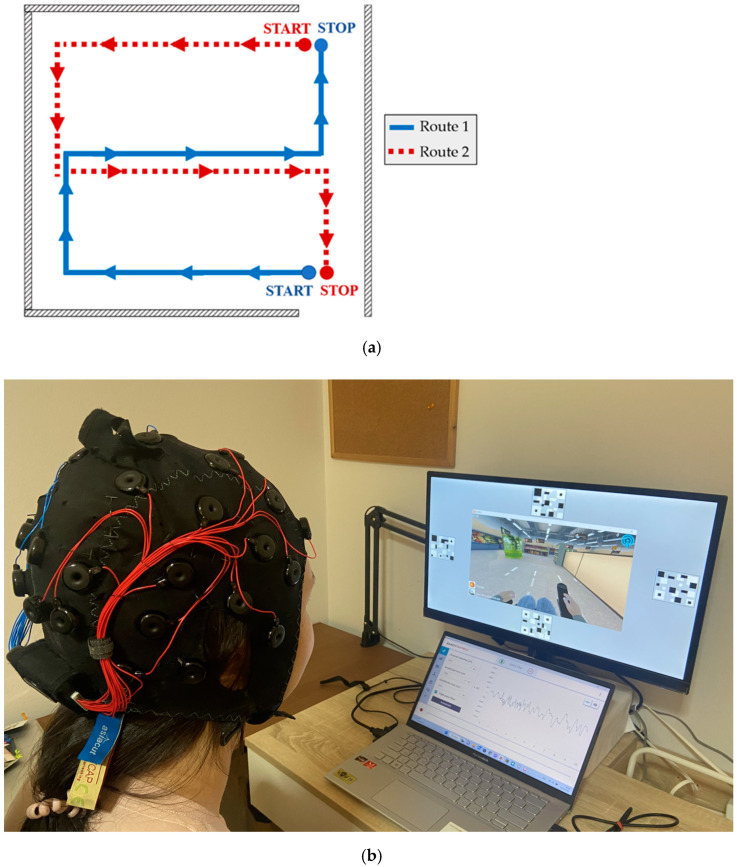
(**a**) The routes for testing (distance: 20 m per route). (**b**) Example scenario of the experiment while participant uses the proposed BCI to control the simulated wheelchair.

**Figure 9 sensors-23-02069-f009:**
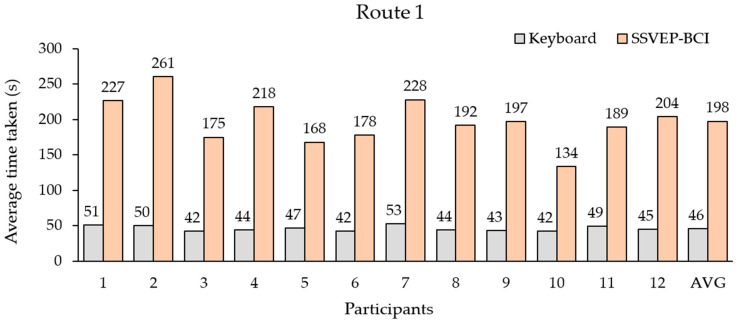
Average times required by all participants to complete route 1.

**Figure 10 sensors-23-02069-f010:**
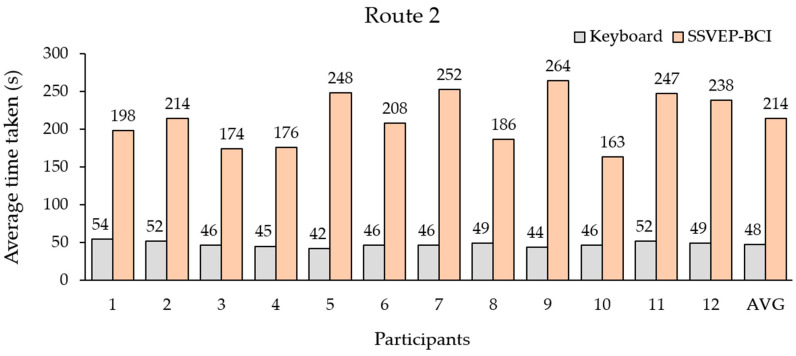
Average times required by all participants to complete route 2.

**Table 1 sensors-23-02069-t001:** Research on EEG-based BCI for wheelchair control.

Authors	BCI Methods	Paradigm and Command	Wheelchair Control	Results
Xiong et al. [33]	MI	Left hand, Right hand, Jaw clench. Imagine left- and right-hand movement and use jaw clench to toggle the wheelchair state.	Actual wheelchair/Assigned control	Accuracy is 60 ± 5% Peak subject accuracy is 82 ± 3%
Permana et al. [34]	Hybrid MI and Eye motion	Think of moving forward and backward and think while moving the eyes. Turning left: think of moving backward while continuously moving the eyes. Turning right: think of moving forward while continuously moving the eyes.	Actual wheelchair/Independent control	Success rate range: 46.67–82.22%
Eidel et al. [35]	Somatosensory P300	Tactile stimulus to right and left thigh, abdomen, and neck.	Virtual wheelchair/Independent control.	Online accuracy 86%
Chen et al. [36]	VEP P300	A translucent visual stimulus panel: visual stimuli. Use a micro-projector to produce flickering visual stimuli (arrow) on the display board.	Actual wheelchair/Assigned control	Accuracy is 88.2%
Yu et al. [37]	Hybrid MI and VEP P300	MI: Left and Right Hand. MI left/right to switch state and to accelerate/decelerate. P300: flashing nine arrow symbols for direction control commands.	Actual wheelchair/Independent control	Average accuracy: MI: 87.2% P300: 92.6%
Chen et al. [38]	SSVEP	Five color images and five flickers (Arrow pattern) with specific frequencies (7, 8, 9, 11, and 13 Hz).	Actual wheelchair/Assigned control	Accuracy in the range of 86.3–98.7%
Na et al. [39]	SSVEP	Combines with LCD and LED Hybrid hardware-driven visual stimulator using Metrix LED for the backlight and LCD for the control panel.	Actual wheelchair/Assigned control	Average accuracy rate 93.93%
Ruhunage et al. [40]	Hybrid SSVEP and EOG	LED Metric for SSVEP stimulus to control wheelchair and change mode to EOG Eyes double blinking for stop command of wheelchair control.	Actual wheelchair/Assigned control	SSVEP accuracy is 84.5% Double blink accuracy is 100%
Punsawad et al. [41]	Hybrid SSVEP and Motion VEP	Motion visual stimulus: a vertical strip moving to the left or right at a frequency of 7 Hz. SSVEP method: specific frequencies (7, 8, 9, 11, and 13 Hz) to control the wheelchair and change the mode to Motion VEP.	Actual wheelchair/Independent control	Average accuracy is 85.6%

**Table 2 sensors-23-02069-t002:** Proposed flickering frequencies and first harmonic for SSVEP stimulus.

SSVEP Stimulus Pattern	Flicker Frequencies	
	Fundamental	Harmonics
1	5 Hz	10 Hz
2	6 Hz	12 Hz
3	7 Hz	14 Hz
4	8 Hz	16 Hz

**Table 3 sensors-23-02069-t003:** Proposed flickering frequencies SSVEP stimulus and output commands.

Commands	The Target of Flicker Frequency	Wheelchair Control
1	5 Hz	Turn Right
2	6 Hz	Forward
3	7 Hz	Backward
4	8 Hz	Turn Left
5	-	Idle

**Table 4 sensors-23-02069-t004:** Command sequence for testing the proposed SSVEP-based BCI for wheelchair control.

Sequence	Wheelchair Control	Sequence	Wheelchair Control
1	Turn Left	5	Turn Right
2	Turn Right	6	Turn Left
3	Forward	7	Backward
4	Backward	8	Forward

**Table 5 sensors-23-02069-t005:** Average classification accuracy of SSVEP feature methods with different SSVEP stimulus patterns of each participant.

SSVEP Features	Average Classification Accuracy (%)							
	Absolute PSD				Relative PSD (Proposed)			
Visual Stimulus Patterns and Flickers	Checkerboard		QR Code		Checkerboard		QR Code	
Participants	Single	Mixture	Single	Mixture	Single	Mixture	Single	Mixture
1	75.0	79.2	70.8	85.4	81.3	85.4	83.3	91.7
2	72.9	79.2	75	81.3	83.3	85.4	85.4	87.5
3	72.9	77.1	70.8	77.1	87.5	85.4	81.3	89.6
4	75.0	81.3	70.8	83.3	83.3	91.7	85.4	95.8
5	70.8	75.0	70.8	79.2	79.2	87.5	81.3	89.6
6	75.0	77.1	77.1	81.3	85.4	89.6	85.4	93.8
7	70.8	72.9	77.1	81.3	81.3	83.3	87.5	95.8
8	72.9	77.1	75	83.3	77.1	79.2	81.3	85.4
9	75.0	81.3	77.1	81.3	83.3	85.4	87.5	91.7
10	75.0	75.0	75	75.0	83.3	85.4	85.4	93.8
11	75.0	77.1	77.1	79.2	81.3	87.5	83.3	93.8
12	72.9	77.1	75	79.2	81.3	87.5	85.4	95.8
Mean ± SD.	73.6 ± 1.62	77.4 ± 2.49	74.3 ± 2.71	80.6 ± 2.86	82.3 ± 2.74	86.1 ± 2.26	84.4 ± 3.12	92.0 ± 3.42
ITR (bpm)	12.7 ± 0.0	13.9 ± 2.8	12.7 ± 0.0	17.0 ± 3.6	18.9 ± 2.8	19.5 ± 2.0	20.1 ± 0.0	26.1 ± 6.0

## Data Availability

The data presented in this study are available upon reasonable request.

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
