# Peer review of "Brain–Computer Interface Based on Steady-State Visual Evoked Potential Using Quick-Response Code Pattern for Wheelchair Control"

_sensors, 2023, doi:10.3390/s23042069_

Round 1
Reviewer 1 Report
This is an excellent addition to the literature. Combining the QR code with relative PSD seems like a useful approach. Well done.
Reviewer 2 Report
The paper presents a Visual Steady-State Based Brain-Computer Interface 2 Evoked Potential Using Quick-Response Code Pattern For 3 Wheelchair Control, this topic is of interest and relevance. In the attached paper there are some comments and questions. In general, the work is well presented, only the samples in the case study (12 people) are few, and it is also recommended to include the age range of the users, since the response can be affected by this variable.

Reviewer 3 Report
The paper is well written and structured, and the conclusions are supported by the presented results. Yet, I still have three major concerns that should be addressed by the authors.
1. I could not find any information on the preprocessing. The authors are expected to clearly write how and what filters (if any) was used for the preprocessing.
2. I expect the authors to add Information Transfer Rate (ITR) index to their study. I believe the ITR shows a good trad-off between the accuracy and required time to run the algorithm. You may use (10.1109/JBHI.2021.3096984) as a potentail reference.
3. The authors are expected to provide a table or graph showing the accuracy per subject . It is of great importance to compare the perfromance of the proposed method on both intra- and inter- subject variability.
Reviewer 4 Report
This manuscript covers brain–computer interface based on steady-state visual evoked potential using quick-response code pattern for wheelchair control, and the respectable authors utilized a quick-16 response (QR) code visual stimulus pattern for a robust existing system. On the whole, the manuscript is interesting and has important prospects for application. The following comments are suggested.
(1) The resolution of Figs. 7 (a) and (b) is low, and some texts in them cannot be clearly displayed, which maybe make reading difficult.
(2) In Table 1, the authors listed some researches on EEG-based BCI for wheelchair control, and some methods have exceeded the methods proposed in this manuscript. However, the authors only have compared the performances of SSVEP-BCI with those of keyboard methods. It is suggested that the authors increase the comparison with recent similar researches and analyze the unique advantages of the proposed method.
(3) In Experiments, please introduce some specific information about the computer adopted additionally, such as CPU, memory frequency, Python version, etc. These factors may affect the performances of the proposed method. It will be conducive to the recurrence of results or the comparison of peers.
(4) Lines 209 and 233 follow the previous paragraph and should be written in the top case, not another paragraph.
(5) What is the meaning of " According to the results in Section 3.2" in line 365? This paragraph is exactly in section 3.2.
Reviewer 5 Report
In this paper, authors proposed the use of relative PSD method to extract features of EEG-based BCI based on SSVEP. The work is interesting, but their is a room for improvement as follows:
1- Please avoid the repetition in defining abbreviations such as ( BCIs in lines 13 and 32, EEG in lines 15 and 38, MI in lines 56 and 62, SSVEP in lines 16 and 59, and PSD in lines[19,19,194, and197])
2- Please don't define abbreviation in figure legend as in figure 7.
3- Authors didn't refer to reference 41 in the body of the paper.
4- In lines 309 and 312, authors mentioned figure 7(b) by mistake. It should be figure 2 a and b.
5- In the text, authors commented figure 2(b) before (a). Please do it alphabetically. Also for figure 7, authors commented 7(b) then (d) then (c), and didn't comment (a) at all.
6- In line 315, authors reported 64 command, please clarify. Also the sequence number in table 4 needs explanation.
7- In line 341 authors used n= 24 for t test and n= 48 (in line 346). Please clarify this issue for reader when sample number (12 participants).
Round 2
Reviewer 3 Report
The authors have addressed all concerns raised by me. I have no further comments.
Reviewer 4 Report
Now I have no further suggestions.